# Chemical Composition of Essential Oils and Supercritical Carbon Dioxide Extracts from *Amomum kravanh*, *Citrus hystrix* and *Piper nigrum* ‘Kampot’

**DOI:** 10.3390/molecules28237748

**Published:** 2023-11-24

**Authors:** Vihanova Katerina, Urbanova Klara, Nguon Samnang, Kokoska Ladislav

**Affiliations:** 1Department of Crop Science and Agroforestry, Faculty of Tropical AgriSciences, Czech University of Life Sciences Prague, Kamycka 129, 165 21 Prague 6-Suchdol, Czech Republic; vihanova@ftz.czu.cz; 2Department of Sustainable Technologies, Faculty of Tropical AgriSciences, Czech University of Life Sciences Prague, Kamycka 129, 165 21 Prague 6-Suchdol, Czech Republic; urbanovak@ftz.czu.cz; 3Graduate School, Royal University of Agriculture, Dangkor, P.O. Box 2696, Phnom Penh 12401, Cambodia; sam@rua.kh

**Keywords:** essential oil, GC-MS analysis, spice, supercritical fluid extraction

## Abstract

The fruits of *Amomum kravanh*, *Citrus hystrix* and *Piper nigrum* ‘Kampot’ are traditionally used as spices in Cambodian cuisine. In this study, the chemical composition of essential oils (EOs) and supercritical CO_2_ extracts from all three species was determined using GC-MS, with two columns of different polarity (HP-5/DB-HeavyWAX). Differences between the chemical profile of the EOs and CO_2_ extracts were observed for all species. The greatest difference was detected in *A. kravanh* EO containing mainly eucalyptol (78.8/72.6%), while the CO_2_ extract was rich in fatty acids (13/55.92%) and long-chain alkanes (25.55/9.54%). Furthermore, the results for the CO_2_ extract of this species differed, where tricosane (14.74%) and oleic acid (29.26%) were the main compounds identified when utilizing the HP-5 or DB-HeavyWAX columns, respectively. Moreover, the EO and CO_2_ extract from *P. nigrum* ‘Kampot’ fruits and the CO_2_ extract from *C. hystrix* fruit peel, containing respective amounts 34.84/39.55% (for EO) and 54.21/55.86% (for CO_2_ extract) of β-caryophyllene and 30.2/28.9% of β-pinene, were isolated and analyzed for the first time. Generally, these findings suggest that supercritical CO_2_ could potentially be used for the extraction of all three spices. Nevertheless, further research determining the most efficient extraction parameters is required before its commercial application.

## 1. Introduction

The term “spice” refers to dried plants, or their parts, that are used to enhance food flavor, taste, and color [1,2]. Nowadays, more than 400 spices and condiments are used worldwide, of which among 275 species have their origin in tropical Southeast Asia. Cardamom, cinnamon, clove, ginger, nutmeg, pepper, and turmeric are examples of commodities of global economic importance. In 2021, the total global spice market values accounted for 21.3 billion US dollars, and it is forecasted to reach 27.4 billion USD by the end of 2026 [3]. Economically, the most important spice in international trade is black pepper, known as the “king” of the spices. Pepper, or peppercorn, refers to the dried fruits of *Piper nigrum*, a perennial vine native to Western Ghats in India, which belongs to the Piperaceae family. Throughout history, black pepper was one of the most-traded spices worldwide and it was even utilized as a currency along the commercial routes established between Europe and India [4,5]. Currently, the global black pepper market is estimated to have reached 4400 million USD in 2022 and is likely to increase to almost 8 million USD by 2032 [6]. Besides their seasoning properties, spices are used as natural colorants in the food industry due to the presence of pigments. Moreover, due to the numerous proven beneficial effects attributed to active biochemicals present in spices, they are also utilized in aromatherapy, cosmetics, nutraceuticals, perfumes, and pharmaceuticals. Spices can be added to foods in various forms, such as whole, ground, or in the form of highly concentrated extracts [7].

In most spices, essential oils (EOs) are the main constituents responsible for their taste and olfactory properties. EOs are usually comprised of a large number of individual constituents (up to 400) with one or two dominant compounds, mostly classified as terpenes and their oxygenated derivatives. Other chemicals present in EOs include phenylpropanoids, which are responsible for the aroma of spices. Significant representatives include phenols or phenolic ethers [8,9,10]. Furthermore, alkaloids also contribute to the olfactory properties of EOs, especially the pungency of some spices [11,12]. The extraction of EOs can be carried out using a wide range of techniques; however, distillation (steam, water, or combined) remains the most common method applied on an industrial scale. Extracts obtained by solvent extraction are called oleoresins and contain flavor constituents of spices and other compounds soluble in organic solvents [2,4,13]. The most important shortcomings of distillation are the high consumption of plant material, loss of thermo-sensitive compounds, and long extraction times. The main disadvantages of solvent extraction are the environmental and safety hazards associated with the accumulation of organic solvents, high energy costs, and the oxidation of aroma and coloring compounds from spices [14].

To overcome these drawbacks, various green extraction techniques have recently been developed. Supercritical fluid extraction (SFE) provides multiple advantages associated with the utilization of supercritical fluids as solvents possessing different physicochemical properties. Their lower viscosity and higher diffusivity result in higher extraction rates and an overall faster process. Moreover, their density, which influences the solvent capacity, can be modified by adjusting the extraction parameters [14,15]. Carbon dioxide (CO_2_) is the most commonly used supercritical fluid due to its wide availability at low cost, and non-toxic and non-flammable properties. Moreover, its temperate critical pressure and temperature can ensure the preservation of labile compounds in the final extract [16,17]. All of the above-mentioned characteristics make supercritical CO_2_ a highly attractive “green solvent”, which has led to multiple practical applications in different industries. Hop extract, decaffeinated coffee, nicotine-free tobacco, and specialty oils are a few examples of commercial products utilizing supercritical CO_2_ extraction at an industrial level [18,19]. Extensive research has considered the possible alternative uses of SFE in the extraction of bioactive components from spices during recent decades and has led to the availability of a wide variety of products on the market, including CO_2_ extracts from *Cinnamomum verum*, *Piper nigrum* and *Zingiber officinale* [17]. Regarding the differences in chemical composition of EOs and supercritical CO_2_ extracts from spices and aromatic plants, a plethora of studies have been conducted. Most commonly, results indicated significant differences in the quantities of individual compounds identified. EOs obtained through distillation contained higher amounts of low-molecular-weight components, like monoterpenoids and phenylpropanoids, while CO_2_ extracts were richer in constituents of higher molecular weight, like sesquiterpenoids and diterpenoids [20,21,22,23].

Cambodia, as well as many Southeast Asian countries, has a cuisine that is generally considered to be healthy and delicious, due to the abundant use of fish and the incorporation of many vegetables, fruits, herbs, and spices into every meal. Fresh spices and herbs are essential ingredients in Cambodian dishes, and approximately 42 g of condiments and spices are consumed per person daily [24,25]. Kampot pepper is one of the most popular spices grown in Cambodia with international recognition. It is a cultivar of *Piper nigrum* L. produced in the Kampot province which has unique climatic and soil conditions, giving the pepper its distinctive flavor and aroma compared with other kinds of peppercorns. This commodity has been exported to Europe since 1870, during the French Protectorate, due to its exceptional organoleptic quality. Currently, four different types of Kampot pepper can be found on the market: green, black, red, and white pepper. Although all Kampot peppers have an excellent reputation regarding their sensory properties, red peppercorns are especially rare because of the unique process of their production. Red berries are harvested in full maturity, then blanched, sun-dried, and manually sorted [26,27].

Leaves and fruits of numerous *Citrus* species (*Rutaceae*) are widely used to flavor foods and beverages. *Citrus hystrix* DC, known as kaffir lime, is an example of a regionally used condiment in Cambodia. Leaves and fruit juice from this citrus are used for various flavoring purposes in Khmer cuisine, and EO from the fruit pericarp is utilized in cosmetics and beauty products. Previous research has shown that kaffir lime EO possesses various biological activities, namely antimicrobial, antioxidant, insect-repellent, and antiviral. Numerous studies analyzing the chemical composition of *C. hystrix* EO have been conducted and revealed the presence of a substantial number of monoterpene hydrocarbons and their oxygenated derivatives, with the main compounds being α/β-pinene, citronellal, limonene and sabinene [28,29,30]. 

Plants belonging to the *Zingiberaceae* family are also widely used as spices in Southeast Asia because of their unique aroma. *Amomum kravanh* Pierre ex Gagnep. (*Zingiberaceae*) is cultivated in Cambodia and other Southeast Asian countries for its fruits and leaves that are used to flavor curries. Fruit EO and SFE extract from this plant contain eucalyptol and β-pinene as their main components [31,32,33,34]. With the exception of a single report on the chemical composition of *C. hystrix* leaf CO_2_ extract [35], there is no study dealing with the supercritical extraction of the fruits from these spice species that are frequently used in Southeast Asian cuisine. Moreover, the supercritical CO_2_ extract from *P. nigrum* has shown a higher antioxidant effect when compared to hydrodistilled EO, and chemical analysis revealed a higher recovery of thermo-labile compounds in the CO_2_ extract [17]. Therefore, the main objective of this study was to determine the chemical composition of EOs and CO_2_ extracts obtained from the fruits of three traditional Cambodian spices, namely *A. kravanh*, *C. hystrix*, and *P. nigrum* ‘Kampot’.

## 2. Results

In this investigation, three EOs and three CO_2_ extracts were isolated from Cambodian spice species, with respective yield values ranging from 3.01 to 5.22% (EOs) and from 0.57 to 8.35% (CO_2_ extracts). In EOs obtained from *P. nigrum* ‘Kampot’, *C. hystrix*, and *A. kravanh*, a total of 35, 38, and 21 individual constituents were identified using the HP-5 column, representing 99.38, 98.68, and 99.88% of their respective total contents. Using the DB-HeavyWAX column, a total of 41, 50, and 24 compounds were detected, constituting 99.15, 98.06, and 99.26% of the total EOs for *P. nigrum* ‘Kampot’, *C. hystrix*, and *A. kravanh*, respectively. In CO_2_ extracts from these same spices, a total number of 32, 36, and 31 components were determined when using the HP-5 column, amounting to 98.65, 99.36, and 92.69% of the total extracts. When utilizing the DB-HeavyWAX column, 40, 54, and 40 compounds were identified, which accounted for 96.74, 98.06, and 95.57% of their total respective contents for *P. nigrum* ‘Kampot’, *C. hystrix*, and *A. kravanh*. Sesquiterpenes, monoterpenes, and their oxygenated derivatives were the most predominant chemical groups in almost all tested EOs and CO_2_ extracts from these three spices, with the exception of the *A. kravanh* extract, where the most abundant chemicals were higher fatty acids and long-chain alkanes. 

In *A. kravanh* EO, oxygenated monoterpene eucalyptol was the prevailing compound, comprising 78.8/72.6% of the total sample. Other compounds occurring in significant amounts were monoterpenes β-pinene (7.68/7.49%), α-pinene (2.3/2.2%) and oxygenated derivative α-terpineol (4.31/4.67%). Following analysis with the HP-5 column, L-terpinene-4-ol amounted to 1.19% of the EO; however, this constituent was not detected by the DB-HeavyWAX column, where monoterpene limonene was the third-most abundant component (5.12%) identified. On the contrary, the chemical composition of the CO_2_ extract differed substantially from the EO. When utilizing the HP-5 column, the long-chain alkane tricosane accounted for 14.74% of the total extract, followed by eugenol acetate which accounted for 14.02% of the extract. Oleic acid was the third-most prevailing constituent, comprising 12.21% of the extract accompanied by phenylpropanoid eugenol (7.91%) and long-chain alkane pentacosane (5.19%). In contrast with these findings, analyses with the DB-HeavyWAX column differed considerably. The majority of the extract consisted of oleic and palmitic acids, constituting 29.26 and 17.07% of the total respective content, followed by tricosane (5.26%), eugenol acetate (5.24%), and linoleic acid (5.17%). Compared to hydrodistillation, the yield of the CO_2_ extract was much lower (0.6%), and its physical properties were different, as the extract had a waxy and semi-solid structure.

Investigation of *C. hystrix* EO revealed that monoterpenes were the most prevalent class of chemical compounds. Monoterpenes β-pinene (29.95/29.45%), limonene (24.54/23.24%), and sabinene (9.94/10.23%), accompanied by alcohols L-terpinene-4-ol (9.71/9.07%) and α-terpineol (3.7/3.62%), were the main constituents of the EO. Similarly, β-pinene (30.2/28.9%), limonene (23.99/23.74%), and sabinene (13.36/19.55%), followed by aldehyde citronellal (5.21/4.28%), were predominant components of the CO_2_ extract. Following HP-5 column analysis, furanocoumarin oxypeucedanin accounted for 2.96% of the total extract; however, this compound was not detected by the DB-HeavyWAX column.

In *P. nigrum* ‘Kampot’ EO, sesquiterpene β-caryophyllene was identified as the dominant compound, constituting 34.84/39.55% of the total oil, followed by monoterpenes 3-carene (18.72/18.48%), limonene (11.18/10.93%), and β-pinene (5.42/5.32%) when utilizing HP-5/DB-HeavyWAX columns, respectively. Similarly, analysis of the CO_2_ extract revealed an even higher content of β-caryophyllene (54.21/55.86%) accompanied by 3-carene, limonene, and β-selinene, comprising 7.4/7.18%, 6.26/6.03%, and 5.24/4.76% of the total extract. Complete chemical analyses of *A. kravanh*, *C. hystrix*, and *P. nigrum* ‘Kampot’ EOs and CO_2_ extracts are provided in Table 1, Table 2 and Table 3. Chromatograms of EOs and CO_2_ extracts can be seen in Figure 1 and Figure 2.

## 3. Discussion

As a result of the GC-MS analysis, eucalyptol was the dominant constituent of *A. kravanh* EO. This finding is in accordance with previously published studies investigating the chemical composition of EO from this plant [31,32], and from other species of the *Amomum* genus [46,47]. In correspondence with the results of Zhang et al. [33], β-pinene and α-terpineol were the abundant compounds in the analyzed sample of the *A. kravanh* EO. Contrastingly, Diao et al. [31] reported relatively lower amounts of α-pinene (5.71%) and β-pinene (2.41%), while terpinyl acetate (11.2%) and dipentene (6.1%) were abundant EO components. These slight differences can be attributed to different geographical origins of the samples. Corresponding with Zhang et al. [33], limonene was identified as the third-most prevalent compound of the EO from *A. kravanh* when utilizing the DB-HeavyWAX column. The current literature only reports the use of the DB-HeavyWAX column, the results of which differ from those reported in the present study. According to Yothipitak et al. [34], eucalyptol (71.45%), β-pinene (8.64%), and limonene (4.77%) were the three dominant constituents of the *A. kravanh* extract obtained by SFE. These discrepancies can be caused by the different extraction parameters (33 °C and 175 bars) used during the SFE process and by the distinct geographical origin (Thailand) of the plant sample. Different main constituents identified during the HP-5 column and DB-HeavyWAX column analyses could be the result of the stronger detection sensitivity and ability of the polar DB-HeavyWAX column to separate and quantify fatty acids and their methylesters from the rest of the sample compared to the non-polar HP-5 column. Furthermore, polar columns based on polyethylene glycol have more accurate results regarding the identification of fatty acid saturation and therefore are commonly employed in the analyses of complex fats and oils [48,49].

In *C. hystrix* EO, β-pinene was the most abundant constituent, which is in accordance with previously published analyses, where the percentages of β-pinene ranged from 25.93 to 47.93% [28,50,51,52]. Jantan et al. [28] and Tran et al. [51] also reported limonene as the second-most dominant compound, comprising almost 15 and 20% of the sample, respectively. Sabinene was the third-most abundant constituent in the present study, which agrees with the above-mentioned study of Tran et al. [51]. However, a slight discrepancy can be observed in comparison to the investigation conducted by Sato et al. [50], where this monoterpene accounted for more than 20% of the extract and was the second-most dominant compound of the total oil. Moreover, the *C. hystrix* extract analyzed in the present study was lower in citronellal in comparison to previously published data. Since the samples from previously published studies were collected in Malaysia [28], Vietnam [53], and Thailand [50], differences in chemical composition can be attributed their different geographical origins. Furthermore, in the case of the study conducted by Sato et al. [50], steam distillation was used as the extraction method. In addition, maturity of the fruit and processing of the sample before extraction are factors which can affect the chemical composition of the EO [29]. Although CO_2_ extraction was previously performed from the leaves of this species by Norkaew et al. [35], to the best of our knowledge, this is the first report investigating the chemical composition of CO_2_ extract isolated from the peel of this species. Due to the existence of large secretory cavities in the *Citrus* spp. fruit rind, their EOs have traditionally been obtained by cold-pressing. Cold-pressed EOs from citruses comprise a volatile fraction with mono and sesqui-terpenes and their oxygenated derivatives. However, a non-volatile fraction represented by coumarins, psoralens, and other oxygen heterocyclic compounds is also present in cold-pressed oils [54,55]. Although investigation of *C. hystrix* cold-pressed EO is currently not available in the literature, several studies compared cold-pressed and hydrodistilled EOs from more common *Citrus* species. The most common conclusion was that there is a higher recovery of terpene hydrocarbons in cold-pressed oils, which are compounds responsible for the typical aroma of *Citrus* oils [53,56]. Therefore, the comparison of cold-pressed *C. hystrix* EO with other extraction methods is highly encouraged for future research related to chemical composition or bioactivity assessment.

The main difference between the *P. nigrum* ‘Kampot’ EO and CO_2_ extract was the presence of pellitorine, belonging to the piperamides, which amounted to more than 1% of the total CO_2_ extract. This nitrogen-containing compound has also been detected by Luca et al. [57] in much lower amounts (0.18%); however, other piperamides like piperine, piperettine, and guineesine were also determined in their extracts. This slight dissimilarity can be attributed to the different *P. nigrum* cultivar assessed in in the present study and the different extraction conditions of the SFE process, where higher pressure (up to 300 bars) was used for the selective recovery of piperazines. The predominant compound in the EO was sesquiterpene β-caryophyllene, which is in consonance with numerous previously published studies assessing the composition of *P. nigrum* EO. This sesquiterpene was present in amounts ranging from 29.9 to 62.3% of the volatile oil [58,59,60]. Other major constituents in the *P. nigrum* EO within the present study were 3-carene and limonene. This corresponds well with Li et al. [59], where 3-carene and limonene were present in maximal respective amounts of 26.84 and 25.83% in the various EOs obtained from black and white peppercorns of Chinese origin. However, slightly different components were discovered by Andriana et al. [58], where β-thuj and β-selinene accounted for 20.58 and 5.59% of the sample, respectively. Furthermore, Kapoor et al. [60] and Bagheri et al. [61] reported limonene (13.2%), β-pinene (7.9%), and sabinene (5.9%) as the predominant compounds of the EO. According to the ISO guideline, EO from black peppercorns should contain 25–26% of β-caryophyllene, followed by limonene (11.5–13.4%), sabinene (9.1–9.6%) and α-pinene (8.4–9.7%) [62]. These slight differences in the main components can be attributed to a different cultivar of *P. nigrum* being assessed in the present study, together with different harvest and post-harvest handling of the fruits used to produce dried red Kampot peppercorns [27]. In the CO_2_ extract, the amount of β-caryophyllene was even higher than in the EO, which is in accordance with Bagheri et al. [61], where the recovery of this sesquiterpene was also higher than in the hydrodistilled EO. Moreover, a higher recovery of sesquiterpenes and their oxygenated derivatives for CO_2_ extracts opposed to EOs has also been previously reported [57,62]. The main components in the *P.nigrum* ‘Kampot’ CO_2_ extract were monoterpenes 3-carene and limonene, and such findings correspond well with Topal et al. [63]. These same compounds amounted to 10.32 and 5.4% in the *P. nigrum* CO_2_ extract, respectively. A slight discrepancy can be observed compared to the results of Luca et al. [57], where sabinene was present in 8.61% and limonene comprised 8.21% of the total *P. nigrum* extract, which can again be attributed to the different cultivar researched in the present study. In addition, to the best of our knowledge, the chemical composition of *P. nigrum* ‘Kampot’ EO and CO_2_ extract has been assessed for the first time in the current report.

## 4. Materials and Methods

### 4.1. Plant Material and Sample Preparation

Fruits of *A. kravanh* and *C. hystrix* were purchased in local markets (Orussey Market, Phnom Penh, KH and Stung Treng Market, Cambodia) and *P. nigrum* ‘Kampot’ fruits (red peppercorns) were obtained in a pepper farm store (La Plantation, Kampot, KH). *C. hystrix* was peeled and the pericarp was used for further analyses. Dried material was homogenized using a Grindomix apparatus (GM 100 Retsch, Haan, Germany). The residual moisture contents of the samples were determined gravimetrically at 130 °C for 1 h using a Scaltec SMO 01 analyzer (Scaltec Instruments, Gottingen, Germany), in triplicate, according to the Official Methods of Analysis of the Association of Official Agricultural Chemists and expressed as arithmetic averages (15.79%, 22.51%, and 14.39% for *A. kravanh*, *C. hystrix* and *P. nigrum*, respectively).

### 4.2. Hydrodistillation of EOs

EOs were extracted by the hydrodistillation of 100 g of ground plant materials in one liter of distilled water for 3 h using a Clevenger-type apparatus (Merci, Brno, Czech Republic). The distillation rate was 2–3 mL of liquid/min according to the procedure described in [64]. Since hydrodistillation belongs to the most utilized methods for the commercial production of EOs from *C. hystrix* spp. [29], the properties of the samples prepared in the current investigation should be alike to those commercially available. All EOs were stored in 2 mL sealed glass vials at 4 °C until further use.

### 4.3. Supercritical CO_2_ Extracts Preparation

Supercritical CO_2_ extraction was carried out using a Spe-ed SFE helix system (Applied Separations, Allentown, PA, USA). Initially, 10 g of ground material was placed into a 100 mL stainless steel extraction vessel between a glass wool bilayer. Subsequently, the filled vessel was installed into the extraction module and the extraction process was carried out using following parameters: isocratic pressure 200 Ba, temperature 40 °C, and flow rate of 5 LPM. The extraction continued until no more CO_2_ extracts were obtained (assessed by visual confirmation) whereas total extraction time ranged from 5 to 12 min. The extracts were captured within 60 mL glass collection vials (Applied Separations, Allentown, PA, USA) and stored in 2 mL sealed glass vials at 4 °C until further utilization.

### 4.4. Gas Chromatography–Mass Spectrometry Analysis (GC-MS)

For the determination of the chemical composition of the EOs and supercritical CO_2_ extracts, GC-MS analysis was performed using a non-polar HP-5 column and a polar DB-Heavy WAX column. Since the EOs and CO_2_ extracts from fruits often contain fatty acids and their methyl esters [41], a polar DB-Wax column with an increased ability to separate these constituents was used together with a non-polar column for the analysis of EOs [41,58]. An Agilent GC-7890B. System was utilized, equipped with an Agilent 7693 auto sampler, two columns: a fused-silica HP-5MS column (30 m × 0.25 mm, film thickness 0.25 μm, Agilent 19091s-433) and a DB-HeavyWAX column (30 m × 0.25 mm, film thickness 0.25 μm, Agilent 122–7132), and a flame ionization detector (FID) coupled with single quadrupole mass selective detector (Agilent MSD-5977B, Agilent Technologies, Santa Clara, CA, USA). Helium was utilized as a carrier gas at a flow rate of 1 mL/min and the injector temperature was set to 250 °C for both columns. The oven temperature was raised for both columns after 3 min, from 50 to 280 °C. Initially, the heating velocity was 3 °C/min, until the system reached 120 °C. Subsequently, the velocity increased to 5 °C/min until a temperature of 250 °C was reached, and after 5 min holding time, the heating speed reached 15 °C/min until 280 °C was finally obtained. Heating was followed by a 20 min isothermal period. Samples of EOs and supercritical CO_2_ extracts were diluted in *n*-hexane for GC–MS (Merck KGaA, Darmstadt, Germany) at a concentration of 20 μL/mL. An amount of 1 μL of the solution was injected in split mode in a split ratio of 1:30. The mass detector was set to the following conditions: ionization energy 70 eV, ion source temperature 230 °C, scan time 1 s, and mass range 40–600 *m*/*z*.

### 4.5. Identification of Constituents, Quantification, and Statistical Analysis

Identification of compounds was based on a comparison of their retention indices (RI), retention time (RT), and mass spectra with those in the National Institute of Standards and Technology Library ver 2.0.f (NIST) as well as in the literature [36,37,38,39,40,41,42,43,44,45]. The certain identified compounds were confirmed by co-injection of authentic standards, namely camphene (97.5%, CAS: 79-92-5), β-caryophyllene (80%, CAS: 87-44-5), α-humulene (96%, CAS: 6753-98-6), linalool (97%, CAS: 78-70-6), α-phellandrene (95%, CAS: 4221-98-1), α-pinene (99%, CAS: 7785-70-8), β-pinene (99.0%, CAS: 18172-67-3), and γ-terpinene (97%, CAS: 99-85-4) (Sigma-Aldrich, Prague, Czech Republic). The RI were calculated for the constituents separated by the HP-5 column using the RT of the *n*-alkanes series ranging from C_8_ to C_40_ (Sigma-Aldrich, Prague, Czech Republic). For each analyzed EO and CO_2_ extract, the final number of individual constituents was computed as the sum of components simultaneously identified using both columns and the remaining compounds detected by individual columns only. Quantitative data are expressed as the relative percentage content of the constituents determined by FID. Chemical analysis of EOs and CO_2_ extracts was performed in triplicate and the relative peak area percentages were expressed as the mean average of these three independent measurements ± standard deviation.

## 5. Conclusions

In summary, this study reports the chemical composition of EOs and CO_2_ extracts from three Cambodian spices, namely *A. kravanh*, *C. hystrix*, and *P. nigrum* ‘Kampot’, using GC-MS equipped with two columns of different polarity. The chemical composition of EOs and CO_2_ extracts from all spice species differed depending on the column utilized. The most significant difference was seen for the EO and CO_2_ extracts from *A. kravanh* fruits. Column choice also influenced the chemical composition of *A. kravanh* fruit CO_2_ extract. When utilizing the non-polar HP-5 column, long-chain alkane tricosane was the main compound identified, while following DB-HeavyWAX analysis, oleic and palmitic acids were the two main constituents of the extract. *C. hystrix* and *P. nigrum* ‘Kampot’ CO_2_ extracts were generally richer in sesquiterpenes and their oxygenated derivatives in comparison to their EOs, where monoterpenes were more abundant. Furthermore, fatty acid derivatives, nitrogen-containing compounds, and other non-volatile constituents were also more prevalent in the CO_2_ extracts. To the best of our knowledge, this study describes the first isolation and chemical analysis of EO and CO_2_ extract from *P. nigrum* ‘Kampot’ fruits, and CO_2_ extract from the peel of *C. hystrix*. These findings suggest that supercritical CO_2_ can potentially be used for the extraction of all three Cambodian spices. Nevertheless, further research determining the most efficient extraction parameters, for example, in connection to desired constituents, is needed before its commercial application in the spice-processing practice.

## Figures and Tables

**Figure 1 molecules-28-07748-f001:**
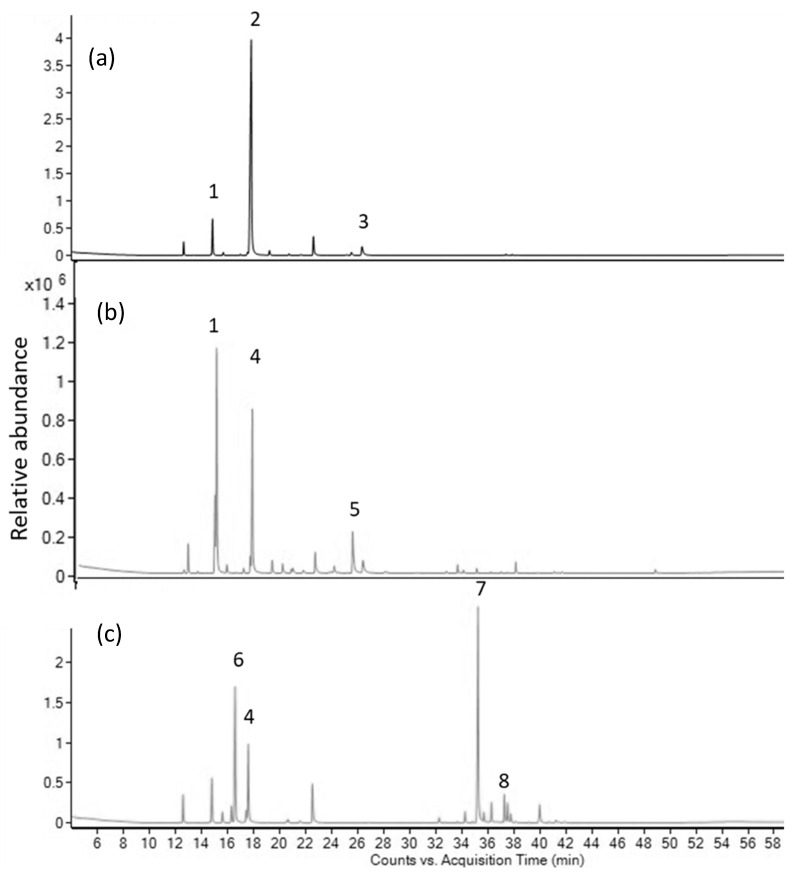
GC-MS chromatograms of EOs from (**a**) *A. kravanh*, (**b**) *C. hystrix* and (**c**) *P. nigrum* ‘Kampot’ (analyzed with HP-5 column). Peak numbers and constituents’ names: 1. β-pinene; 2. eucalyptol; 3. α-terpineol; 4. limonene; 5. terpinene-4-ol; 6. 3-carene; 7. β-caryophyllene; and 8. Β-selinene.

**Figure 2 molecules-28-07748-f002:**
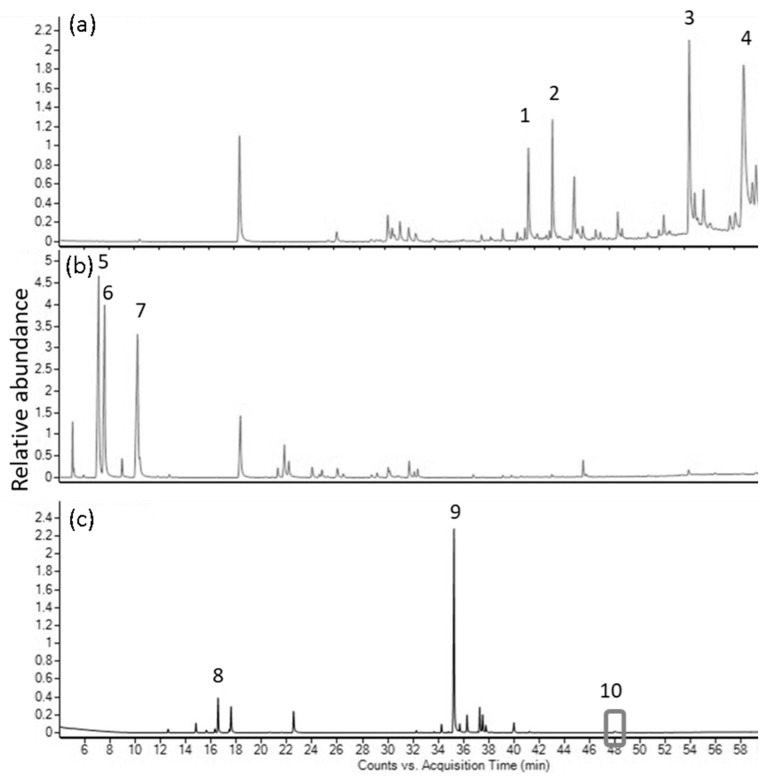
GC-MS chromatograms of CO_2_ extracts from (**a**) *A. kravanh*, (**b**) *C. hystrix* (analyzed on DB-HeavyWax column) and (**c**) *P. nigrum* ‘Kampot’ (analyzed with HP-5 column). Peak number and compound names: 1. Eugenol acetate; 2. Tricosane; 3. Palmitic acid; 4. Oleic acid; 5. Β-pinene; 6. Sabinene; 7. Limonene; 8. 3-carene; 9. Β-caryophyllene; and 10. pellitorine.

**Table 1 molecules-28-07748-t001:** Chemical composition of *A. kravanh* EO and CO_2_ extract.

RI		Compound	C	Extraction Type/Column Type/Peak Area (%)												Column Type/Identification Method	
				Essential Oil						CO_2_ Extract							
Obs.	Lit.			HP-5 MS			DB-Wax			HP-5 MS			DB-Wax			HP-5 MS	DB-Wax
923	931	α-Thujene	MH	0.09	±	0.01	-	-	-	-	-	-	-	-	-	RI, GC-MS	-
929	932	α-Pinene	MH	2.3	±	0.05	2.20	±	0.03	-	-	-	-	-	-	RI, GC-MS, Std	GC-MS, Std
-	945	α-Fenchene	MH	-	-	-	-	-	-	-	-	-	0.01	±	0.01	-	GC-MS
944	946	Camphene	MH	0.07	±	0.01	0.08	±	0.00	-	-	-	-	-	-	RI, GC-MS, Std	GC-MS, Std
970	969	Sabinene	MH	0.2	±	0.04	0.23	±	0.00	-	-	-	-	-	-	RI, GC-MS	GC-MS
973	974	β-Pinene	MH	7.68	±	0.08	7.49	±	0.09	-	-	-	-	-	-	RI, GC-MS, Std	GC-MS, Std
989	988	β-Myrcene	MH	0.78	±	0.03	0.86	±	0.08	-	-	-	-	-	-	RI, GC-MS	GC-MS
1003	1002	α-Phellandrene	MH	0.08	±	0.01	-	-	-	-	-	-	-	-	-	RI, GC-MS, Std	GC-MS, Std
1015	1009	4-Carene	MH	0.24	±	0.03	0.17	±	0.12	-	-	-	-	-	-	RI, GC-MS	GC-MS
-	1014	α-Terpinene	MH	-	-	-	0.22	±	0.01	-	-	-	-	-	-	-	GC-MS
1025	1024	*p*-Cymene	MH	0.69	±	0.04	0.87	±	0.02	-	-	-	-	-	-	RI, GC-MS	GC-MS
1031	1026	Eucalyptol	MO	78.89	±	0.42	72.60	±	0.89	-	-	-	0.08	±	0.01	RI, GC-MS	GC-MS
-	1031	Limonene	MH	-	-	-	5.12	±	0.09	-	-	-	0.01	±	0.01	-	GC-MS
1058	1054	γ-Terpinene	MH	1.05	±	0.06	1.06	±	0.02	-	-	-	-	-	-	RI, GC-MS	GC-MS
-	1083	Fenchone	MO	-	-	-	0.18	±	0.02	-	-	-	-	-	-	-	GC-MS
1087	1086	Isoterpinolene	MH	0.43	±	0.02	-	-	-	-	-	-	-	-	-	GC-MS	-
1105	1095	Linalool	MO	0.45	±	0.01	0.50	±	0.01	-	-	-	-	-	-	RI, GC-MS, Std	GC-MS, Std
1174	1162	δ-Terpineol	MO	0.39	±	0.05	0.43	±	0.01	-	-	-	-	-	-	RI, GC-MS	GC-MS
1182	1174	Terpinen-4-ol	MO	1.19	±	0.12	1.32	±	0.02	-	-	-	-	-	-	RI, GC-MS	GC-MS
1196	1186	α-Terpineol	MO	4.31	±	0.17	4.67	±	0.08	3.68	±	0.16	1.77	±	0.15	RI, GC-MS	GC-MS
1350	1346	α-Terpinyl acetate	MO	-	-	-	-	-	-	0.17	±	0.01	-	-	-	RI, GC-MS	-
1368	1356	Eugenol	PP	-	-	-	-	-	-	7.91	±	0.18	5.06	±	0.18	RI, GC-MS	GC-MS
-	1416	α-Santalene	SH	-	-	-	-	-	-	-		-	0.08	±	0.01	-	GC-MS
1421	1419	β-Caryophyllene	SH	-	-	-	-	-	-	1.37	±	0.25	0.66	±	0.02	RI, GC-MS, Std	GC-MS, Std
1457	1452	α-Humulene	SH	-	-	-	-	-	-	0.41	±	0.03	0.12	±	0.10	RI, GC-MS, Std	GC-MS, Std
1486	1465	(*Z*)-muurola-4(14),5-diene	SH	0.16	±	0.01	-	-	-	-	-	-	-	-	-	RI, GC-MS	-
-	1478	γ-Muurolene	SH	-	-	-	0.16	±	0.02	-	-	-	-	-	-	-	GC-MS
1484	1484	Germacrene D	SH	-	-	-	-	-	-	1.38	±	0.02	-	-	-	RI, GC-MS	-
1490	1489	β-Selinene	SH	0.35	±	0.03	0.25	±	0.01	2.06	±	0.04	0.88	±	0.16	RI, GC-MS	GC-MS
1497	1496	Valencene	SH	-	-	-	-	-	-	0.69	±	0.10	0.09	±	0.01	RI, GC-MS	GC-MS
1508	1505	β-Bisabolene	SH	0.36	±	0.03	0.23	±	0.01	3.59	±	0.06	1.12	±	0.10	RI, GC-MS	GC-MS
1518	1513	γ-Cadinene	SH	-	-	-	-	-	-	1.34	±	0.09	0.73	±	0.11	RI, GC-MS	GC-MS
-	1514	Cubebol	SO	-	-	-	-	-	-	-	-	-	0.39	±	0.01	-	GC-MS
1525	1521	β-Sesquiphellandrene	SH	0.12	±	0.02	0.15	±	0.00	1.60	±	0.62	0.65	±	0.02	RI, GC-MS	GC-MS
1531	1521	Eugenol acetate	PP	-	-	-	-	-	-	14.02	±	0.74	5.23	±	0.11	RI, GC-MS	GC-MS
1558	1542	(*Z*)-Sesquisabinene hydrate	SO	0.12	±	0.02	0.15	±	0.00	0.51	±	0.11	0.20	±	0.01	RI, GC-MS	GC-MS
1566	1561	(*E*)-Nerolidol	SO	-	-	-	0.12	±	0.00	1.71	±	0.03	0.61	±	0.04	RI, GC-MS	GC-MS
1595	1577	(*E*)-Sesquisabinene hydrate	SO	-	-	-	-	-	-	1.11	±	0.30	0.44	±	0.01	RI, GC-MS	GC-MS
-	1577	Spathulenol	SO	-	-	-	-	-	-	-	-	-	0.15	±	0.00	-	GC-MS
1591	1582	Caryophyllene oxide	SO	-	-	-	-	-	-	0.39	±	0.10	0.25	±	0.02	RI, GC-MS	GC-MS
1675	1674	β-Bisabolol	SO	0.04	±	0.04	0.12	±	0.04	0.21	±	0.00	0.07	±	0.00	RI, GC-MS	GC-MS
1691	1685	α-Bisabolol	SO	-	-	-	-	-	-	0.21	±	0.00	0.07	±	0.00	RI, GC-MS	GC-MS
1714	1715	β-Santalol	SO	-	-	-	-	-	-	0.56	±	0.04	0.90	±	0.01	RI, GC-MS	GC-MS
-	1959	Palmitic acid	FAD	-	-	-	-	-	-	-	-	-	17.07	±	0.23	-	GC-MS
2086	2100	Heneicosane	AH	-	-	-	-	-	-	2.12	±	0.03	0.76	±	0.00	RI, GC-MS	GC-MS
-	2113	Linoleic acid	FAD	-	-	-	-	-	-	-		-	5.17	±	0.36	-	GC-MS
2166	2141	Oleic Acid	FAD	-	-	-	-	-	-	12.21	±	0.25	29.26	±	0.42	RI, GC-MS	GC-MS
-	2172	Stearic acid	FAD	-	-	-	-	-	-	-	-	-	2.16	±	0.09	-	GC-MS
2186	2200	Docosane	AH	-	-	-	-	-	-	1.57	±	0.09	0.51	±	0.10	RI, GC-MS	GC-MS
2286	2300	Tricosane	AH	-	-	-	-	-	-	14.74	±	0.60	5.24	±	0.09	RI, GC-MS	GC-MS
2383	2400	Tetracosane	AH	-	-	-	-	-	-	1.94	±	0.03	0.69	±	0.05	RI, GC-MS	GC-MS
2482	2500	Pentacosane	AH	-	-	-	-	-	-	5.19	±	0.39	1.88	±	0.08	RI, GC-MS	GC-MS
-	2700	Heptacosane	AH	-	-	-	-	-	-	-	-	-	0.46	±	0.02	-	GC-MS
2096	NA	m-cymene	K	-	-	-	-	-	-	0.54	±	0.02	-	-	-	GC-MS	-
2261	NA	Tetradec-9-enal	AL	-	-	-	-	-	-	0.24	±	0.01	-	-	-	GC-MS	-
2267	NA	Palmitoleic acid	FAD	-	-	-	-	-	-	0.80	±	0.06	1.94	±	0.03	GC-MS	GC-MS
2464	NA	Hexadec-7-enal	AL	-	-	-	-	-	-	1.50	±	0.13	0.11	±	0.02	GC-MS	GC-MS
2691	2669 ^a^	Azelaic acid bis(2-ethylhexyl) ester	E	-	-	-	-	-	-	4.99	±	0.12	1.96	±	0.20	GC-MS	GC-MS
2810	NA	β-Monoolein	E	-	-	-	-	-	-	2.88	±	0.23	-	-	-	GC-MS	-
-	2478 ^b^	Tricosanol	A	-	-	-	-	-	-	-	-	-	0.38	±	0.01	-	GC-MS
-	NA	Cyclopentadecanone	K	-	-	-	-	-	-	-	-	-	0.46	±	0.01	-	GC-MS
-	2483 ^c^	Pentacos-1-ene	AH	-	-	-	-	-	-	-	-	-	0.48	±	0.04	-	GC-MS
-	2684 ^c^	Heptacos-1-ene	AH	-	-	-	-	-	-	-	-	-	1.31	±	0.04	-	GC-MS
-	1942 ^d^	Hexadec-9-enoic acid	FAD	-	-	-	-	-	-	-	-	-	0.30	±	0.04	-	GC-MS
-	3110 ^c^	Octacosanol	AL	-	-	-	-	-	-	-	-	-	3.16	±	0.06	-	GC-MS
-	NA	Squalene	TH	-	-	-	-	-	-	-	-	-	1.17	±	0.05	-	GC-MS
-	NA	Glyceryl linolenate	E	-	-	-	-	-	-	-	-	-	1.43	±	0.06	-	GC-MS
-	3203 ^e^	β-Sitosterol	O	-	-	-	0.17	±	0.14	-	-	-	-	-	-	-	GC-MS
		Total identified [%]		99.88	±	0.05	99.29	±	0.16	91.64	±	0.88	95.41	±	0.46		

^a^ RI = retention indices for HP-5 column; Obs = retention indices determined relative to a homologous series of *n*-alkanes (C_8_–C_40_) on a HP-5MS column; Lit = literature RI values (Adams, 2007), ^a^ [36], ^b^ [37], ^c^ [38], ^d^ [39], ^e^ [40]; NA = RI values were not available in the literature; ^b^ C = Class. A—Aldehydes; AH—Aliphatic hydrocarbons; AL—Alcohols; E—Esters; FAD—Fatty acid and fatty acid derivatives; K—Ketones; MH—Monoterpene hydrocarbons; MO—Oxygenated monoterpenes; O—Others; SH—Sesquiterpene hydrocarbons; SO—Oxygenated sesquiterpenes; TH = Triterpene hydrocarbons; ^d^ Identification method: GC-MS = Mass spectrum was identical to that of the National Institute of Standards and Technology Library (ver. 2.0.f), RI = the retention index was matching the literature database; Std = constituent identity confirmed by co-injection of authentic standards; ^e^ Retention indices were not calculated for compounds calculated only by the DB-HeavyWAX column.

**Table 2 molecules-28-07748-t002:** Chemical composition of *C. hystrix* EO and CO_2_ extract.

RI		Compound	C	Extraction Type/Column Type/Peak Area [%]												Column Type/Identification Method	
				Essential Oil						CO_2_ Extract							
Obs.	Lit.			HP-5 MS			DB-Wax			HP-5 MS			DB-Wax			HP-5 MS	DB-Wax
923	931	α-Thujene	MH	0.31	±	0.01	0.34	±	0.02	0.32	±	0.02	0.31	±	0	RI, GC-MS	GC-MS
930	937	α-Pinene	MH	2.93	±	0.16	2.96	±	0.09	2.54	±	0.25	2.52	±	0.09	RI, GC-MS, Std	GC-MS, Std
-	945	Fenchene	MH	-	-	-	-	-	-	-	-	-	0.02	±	0.02	-	GC-MS
945	946	Camphene	MH	0.16	±	0.01	0.19	±	0.01	0.14	±	0.01	0.15	±	0	RI, GC-MS, Std	GC-MS, Std
971	976	Sabinene	MH	9.94	±	0.22	10.2	±	0.15	19.36	±	0.97	19.55	±	0.33	RI, GC-MS	GC-MS
974	980	β-Pinene	MH	29.95	±	0.54	29.5	±	0.45	30.2	±	1.84	28.9	±	0.55	RI, GC-MS, Std	GC-MS, Std
990	991	β-Myrcene	MH	1.20	±	0.05	1.38	±	0.02	1.43	±	0.06	1.58	±	0.02	RI, GC-MS	GC-MS
-	1004	Pseudolimonene	MH	-	-	-	0.97	±	0.07	-	-	-	0.97	±	0.04	-	GC-MS
1003	1005	α-Phellandrene	MH	0.07	±	0.01	0.06	±	0	-	-	-	-	-	-	RI, GC-MS, Std	GC-MS, Std
-	1008	3-Carene	MH	-	-	-	0.01	±	0.02	-	-	-	-	-	-	-	GC-MS
1015	1009	4-Carene	MH	-	-	-	-	-	-	0.02	±	0.02	-	±	-	RI, GC-MS	-
1015	1009	α-Terpinene	MH	0.51	±	0.04	0.61	±	0.01	-	-	-	-	-	-	RI, GC-MS	GC-MS
1025	1024	*p*-Cymene	MH	1.91	±	0.01	2.34	±	0.01	0.1	±	0.01	0.21	±	0.02	RI, GC-MS	GC-MS
1028	1031	Limonene	MH	24.54	±	0.16	23.2	±	0.1	23.99	±	0.45	23.74	±	0.12	RI, GC-MS	GC-MS
1058	1062	γ-Terpinene	MH	1.67	±	0.01	1.63	±	0.02	0.08	±	0	0.1	±	0.01	RI, GC-MS	GC-MS
1072	1065	(*Z*)-Sabinene hydrate	MO	-	-	-	-	-	-	0.78	±	0.06	1.06	±	0.02	RI, GC-MS	GC-MS
-	1071	β-Terpinene	MH	-	-	-	-	-	-	-	-	-	0.03	±	0.01	-	GC-MS
1074	1074	Linalool oxide	MO	1.55	±	0.02	1.57	±	0.03	-	-	-	0.05	±	0.01	RI, GC-MS	GC-MS
1087	1086	Terpinolene	MH	0.54	±	0.03	0.52	±	0.01	0.02	±	0.01	0.05	±	0	RI, GC-MS	GC-MS
1105	1095	Linalool	MO	0.72	±	0.09	0.98	±	0.02	0.52	±	0.27	0.81	±	0.07	RI, GC-MS, Std	GC-MS, Std
-	1098	(*E*)-Sabinene hydrate	MO	-	-	-	-	-	-	-	-	-	0.35	±	0.03	-	GC-MS
-	1114	Fenchol	MO	-	-	-	0.02	±	0.03	-	-	-	-	-	-	-	GC-MS
1145	1137	Sabinol	MO	-	-	-	-	-	-	0.08	±	0.02	-	±	-	RI, GC-MS	-
1150	1145	L-isopulegol	MO	0.22	±	0.04	0.28	±	0.03	0.07	±	0.01	0.11	±	0	RI, GC-MS	GC-MS
1154	1148	Citronellal	MO	1.38	±	0.07	1.05	±	0.04	5.21	±	0.31	4.28	±	0.25	RI, GC-MS	GC-MS
1174	1165	Borneol	MO	0.09	±	0.02	-	-	-	-	-	-	-	-	-	RI, GC-MS	-
1161	1167	dl-Isopulegol	MO	0.12	±	0.02	-	-	-	-	-	-	-	-	-	RI, GC-MS	-
1184	1174	Terpinen-4-ol	MO	9.71	±	0.16	9.07	±	0.22	0.36	±	0.04	0.42	±	0.02	RI, GC-MS	GC-MS
-	1182	Pinocarveol	MO	-	-	-	0.03	±	0.03	-	-	-	0.06	±	0	-	GC-MS
-	1183	*p*-Cymen-8-ol	MO	-	-	-	0.09	±	0	-	-	-	-	-	-	-	GC-MS
1199	1189	α-Terpineol	MO	3.70	±	0.09	3.62	±	0.07	0.98	±	0.17	1	±	0.04	RI, GC-MS	GC-MS
-	1194	Myrtenol	MO	-	-	-	0.05	±	0	-	-	-	-	-	-	-	GC-MS
1217	1205	(*E*)-Piperitol	MO	0.02	±	0.04	-	-	-	-	-	-	-	-	-	RI, GC-MS	-
1237	1228	Citronellol	MO	0.75	±	0.15	0.94	±	0.03	0.43	±	0.28	0.92	±	0.04	RI, GC-MS	GC-MS
-	1249	Geraniol	MO	-	-	-	0.1	±	0	-	-	-	-	-	-	-	GC-MS
1291	1273	(*Z*)-Ascaridole glycol	MO	0.40	±	0.10	0.69	±	0.02	-	-	-	-	-	-	RI, GC-MS	GC-MS
-	1312	Citronellic acid	MO	-	-	-	0.3	±	0.01	-	-	-	0.25	±	0.01	-	GC-MS
1351	1345	α-Cubebene	SH	-	-	-	-	-	-	0.05	±	0.02	0.07	±	0	RI, GC-MS	GC-MS
1355	1354	Citronellyl acetate	MO	0.27	±	0.03	0.32	±	0.01	0.33	±	0.11	0.44	±	0.03	RI, GC-MS	GC-MS
1384	1365	Neryl acetate	MO	0.24	±	0.04	-	-	-	0.22	±	0.14	0.01	±	0.02	RI, GC-MS	GC-MS
1379	1374	α-Copaene	SH	0.95	±	0.03	0.75	±	0.04	1.43	±	0.13	1.21	±	0.12	RI, GC-MS	GC-MS
-	1379	Geranyl acetate	MO	-	-	-	0.36	±	0.02	-	-	-	0.51	±	0.03	-	GC-MS
1391	1390	β-Cubebene	SH	0.37	±	0.01	-	-	-	1.04	±	0.15	-	±	-	RI, GC-MS	-
1395	1391	β-Elemene	SH	0.05	±	0.00	-	-	-	-	-	-	-	-	-	RI, GC-MS	-
1424	1419	β-Caryophyllene	SH	0.72	±	0.02	0.57	±	0.01	1.12	±	0.09	1.05	±	0.09	RI, GC-MS, Std	GC-MS, Std
-	1430	β-Copaene	SH	-	-	-	0.31	±	0.01	-	-	-	0.96	±	0.09	-	GC-MS
1461	1454	α-Humulene	SH	0.23	±	0.01	0.2	±	0.01	0.35	±	0.03	0.32	±	0.02	RI, GC-MS, Std	GC-MS, Std
1488	1484	Germacrene D	SH	0.24	±	0.01	0.07	±	0	0.69	±	0.05	0.49	±	0.06	RI, GC-MS	GC-MS
1502	1495	Bicyclogermacrene	SH	-	-	-	-	-	-	0.14	±	0.07	0.13	±	0.01	RI, GC-MS	GC-MS
1505	1499	α-Muurolene	SH	0.10	±	0.01	0.05	±	0.02	0.1	±	0.05	0.09	±	0.01	RI, GC-MS	GC-MS
-	1514	Cubebol	SO	-	-	-	-	-	-	-	-	-	0.24	±	0.02	-	GC-MS
1528	1524	β-Cadinene	SH	1.49	±	0.04	1.06	±	0.1	1.75	±	0.11	1.57	±	0.07	RI, GC-MS	GC-MS
-	1528	Calamenene	SH	-	-	-	0.01	±	0.01	-	-	-	-	-	-	-	GC-MS
-	1548	Elemol	SO	-	-	-	0.09	±	0.02	-	-	-	0.01	±	0.01	-	GC-MS
1588	1574	Germacrene D-4-ol	SO	-	-	-	-	-	-	0.07	±	0.01	0.17	±	0.01	RI, GC-MS	GC-MS
-	1577	Spathulenol	SO	-	-	-	0.09	±	0	-	-	-	0.09	±	0.01	-	GC-MS
-	1582	Caryophyllene oxide	SO	-	-	-	-	-	-	-	-	-	0.05	±	0	-	GC-MS
-	1608	Humulene epoxide	SO	-	-	-	0.01	±	0.01	-	-	-	-	-	-	-	GC-MS
-	1619	Humulane-16-dien-3-ol	SO	-	-	-	-	-	-	-	-	-	0.04	±	0	-	GC-MS
1641	1627	Epicubenol	SO	0.08	±	0.01	0.08	±	0.02	-	-	-	-	-	-	RI, GC-MS	GC-MS
1647	1631	γ-Eudesmol	SO	0.26	±	0.01	0.41	±	0.03	-	-	-	-	-	-	RI, GC-MS	GC-MS
-	1640	τ-Muurolol	SO	-	-	-	0.05	±	0.01	-	-	-	-	-	-	-	GC-MS
-	1645	Cubenol	SO	-	-	-	0.07	±	0.02	-	-	-	-	-	-	-	GC-MS
1656	1645	δ-Cadinol	SO	0.10	±	0.00	0	±	0.01	-	-	-	0.02	±	0.02	RI, GC-MS	GC-MS
-	1649	β-Selinenol	SO	-	-	-	0.18	±	0.02	-	-	-	0.04	±	0	-	GC-MS
1671	1652	α-Eudesmol	SO	0.41	±	0.01	0.06	±	0.01	-	-	-	-	-	-	RI, GC-MS	GC-MS
-	1656	Patchouli alcohol	SO	-	-	-	-	-	-	-	-	-	0.04	±	0.01	-	GC-MS
-	1949 ^d^	Isophytol	DO	-	-	-	-	-	-	-	-	-	0.06	±	0	-	GC-MS
-	1984	Palmitic acid	FAD	-	-	-	-	-	-	-	-	-	0.62	±	0.01	-	GC-MS
-	2132	Linoleic acid	FAD	-	-	-	-	-	-	-	-	-	0.28	±	0.03	-	GC-MS
2521	2501	Oxypeucedanin	O	-	-	-	-	-	-	2.96	±	0.64	-	±	-	RI, GC-MS	-
2707	2707	β-Monolinolein	E	-	-	-	-	-	-	0.18	±	0.01	-	±	-	RI, GC-MS	-
1562	1559 ^b^	Hedycaryol ^b^	SO	-	-	-	-	-	-	0.17	±	0.01	0.17	±	0.02	GC-MS	GC-MS
2009	2032 ^b^	Thunbergol	DO	-	-	-	-	-	-	1.79	±	0.12	1.4	±	0	GC-MS	GC-MS
2010	NA	*(E)*-Geranylgeraniol	DO	0.79	±	0.02	0.64	±	0.01	0.31	±	0.03	0.19	±	0	GC-MS	GC-MS
-	1126 ^a^	2-p-Menthen-1-ol	MO	-	-	-	0.15	±	0.01	-	-	-	-	-	-	-	GC-MS
-	1765 ^c^	Tetradecanoic acid	FAD	-	-	-	-	-	-	-	-	-	0.16	±	0.01	-	GC-MS
-	2199 ^e^	17-Octadecynoic acid	FAD	-	-	-	-	-	-	-	-	-	0.12	±	0.08	-	GC-MS
-	2351 ^a^	Ricinoleic acid	FAD	-	-	-	-	-	-	-	-	-	0.06	±	0.01	-	GC-MS
		Total identified (%)		98.69	±	0.22	98	±	0.39	99.36	±	0.07	98.06	±	0.32		

RI = retention indices for HP-5 column; Obs = retention indices determined relative to a homologous series of n-alkanes (C_8_–C_40_) on a HP-5MS column; Lit = literature RI values [41], ^a^ [38] ^b^ (Liu et al., 2006) [42], ^c^ (Roussis et al., 2000) [43], ^d^ (Palic et al., 2002) [44], ^e^ (Treytakov 2007) [45]; NA = RI values were not available in the literature, C = Class; A—Aldehydes; DO—Oxygenated diterpenes; E—Esters; FAD—Fatty acid and fatty acid derivatives; MH—Monoterpene hydrocarbons; MO—Oxygenated monoterpenes; O—Others; SH—Sesquiterpene hydrocarbons; SO—Oxygenated sesquiterpenes; Identification method: GC-MS = Mass spectrum was identical to that of National Institute of Standards and Technology Library (ver. 2.0.f); RI = the retention index was matching the literature database; Std = constituent identity confirmed by co-injection of authentic standards; ^e^—Retention indices were not calculated for compounds calculated only by the DB-HeavyWAX column.

**Table 3 molecules-28-07748-t003:** Chemical composition of *P. nigrum* ‘Kampot’ EO and CO_2_ extract.

RI		Compound	C ^c^	Extraction Type/Column Type/Peak Area (%)												Identification	
				Essential Oil						CO_2_ Extract							
Obs.	Lit.			HP-5 MS			DB-Wax			HP-5 MS			DB-Wax			HP-5 MS	DB-Wax
923	924	α-Thujene	MH	0.059	±	0	-	-	-	-	-	-	-	-	-	RI, GC-MS	-
929	937	α-Pinene	MH	2.806	±	0.25	2.568	±	0.09	0.649	±	0.02	0.574	±	0.04	RI, GC-MS, Std	GC-MS, Std
944	946	Camphene	MH	0.04	±	0.01	0.05	±	0	-	-	-	-	-	-	RI, GC-MS, Std	GC-MS, Std
970	976	Sabinene	MH	0.048	±	0.03	0.091	±	0	-	-	-	-	-	-	RI, GC-MS	GC-MS
973	980	β-Pinene	MH	5.424	±	0.45	5.322	±	0.14	2.039	±	0.04	1.996	±	0.14	RI, GC-MS, Std	GC-MS, Std
989	991	β-Myrcene	MH	1.477	±	0.14	1.682	±	0.12	0.681	±	0.05	-	-	-	RI, GC-MS	GC-MS
-	1001	2-Carene	MH	-	-	-	-	-	-	-	-	-	0.076	±	0.02	-	GC-MS
1003	1005	α-Phellandrene	MH	1.803	±	0.14	1.481	±	0.08	0.762	±	0.03	0.681	±	0.03	RI, GC-MS, Std	GC-MS, Std
1008	1008	3-Carene	MH	18.72	±	1.46	18.49	±	0.42	7.395	±	0.17	7.181	±	0.4	RI, GC-MS	GC-MS
1025	1024	*p*-Cymene	MH	1.399	±	0.12	1.495	±	0.04	0.636	±	0.01	0.771	±	0.05	RI, GC-MS	GC-MS
1028	1031	Limonene	MH	11.18	±	0.79	10.93	±	0.15	6.265	±	0.11	6.034	±	0.39	RI, GC-MS	GC-MS
1058	1062	γ-Terpinene	MH	0.056	±	0.01	0.045	±	0	-	-	-	-	-	-	RI, GC-MS	GC-MS
1084	1086	Isoterpinolene	MH	0.194	±	0.04	0.4	±	0.02	0.09	±	0	-	-	-	RI, GC-MS	GC-MS
1087	1086	Terpinolene	MH	0.428	±	0.08	0.169	±	0	0.156	±	0.01	0.191	±	0.01	RI, GC-MS	GC-MS
1104	1095	Linalool	MO	0.354	±	0.03	0.453	±	0	0.238	±	0.05	0.386	±	0.01	RI, GC-MS, Std	GC-MS, Std
-	1140	Verbenol	MO	-	-	-	0.185	±	0.02	-	-	-	-	-	-	-	GC-MS
-	1179	p-Cymen-8-ol	MO	-	-	-	0.05	±	0.05	-	-	-	-	-	-	-	GC-MS
-	1318	2,3-Pinanediol	MO	-	-	-	0.254	±	0.01	-	-	-	-	-	-	-	GC-MS
-	1329	Piperonal	O	-	-	-	-	-	-	-	-	-	0.04	±	0.01	-	GC-MS
1339	1339	δ-EIemene	SH	0.559	±	0.02	0.588	±	0.01	0.491	±	0.01	0.516	±	0.01	RI, GC-MS	GC-MS
-	1340	Piperitenone	MO	-	-	-	0.063	±	0.05	-	-	-	-	-	-	-	GC-MS
1351	1351	α-Cubebene	SH	0.097	±	0	0.086	±	0.02	0.144	±	0.01	0.117	±	0.01	RI, GC-MS	GC-MS
-	1357	Octadecanal	A	-	-	-	0.57	±	0.1	-	-	-	-	-	-	-	GC-MS
1378	1374	α-Copaene	SH	0.194	±	0.01	0.17	±	0.01	0.275	±	0.01	0.247	±	0.02	RI, GC-MS	GC-MS
1394	1391	β-Elemene	SH	1.483	±	0.08	-	-	-	1.887	±	0.03	1.303	±	0.03	RI, GC-MS	-
1410	1409	α-Gurjunene	SH	0.164	±	0.01	0.131	±	0	0.257	±	0.01	0.239	±	0.03	RI, GC-MS	GC-MS
1416	1411	α-Bergamotene	SH	0.093	±	0	0.01	±	0.01	0.154	±	0.01	0.018	±	0	RI, GC-MS	GC-MS
1425	1419	β-Caryophyllene	SH	37.84	±	2.05	39.55	±	1.12	54.21	±	0.85	55.86	±	1.37	RI, GC-MS, Std	GC-MS, Std
-	1434	γ-Elemene	SH	-	-	-	0.057	±	0	-	-	-	0.12	±	0.01	-	GC-MS,
1440	1437	α-Guaiene	SH	0.983	±	0.07	-	-	-	1.363	±	0.02	-	-	-	RI, GC-MS	-
1456	1454	β-Farnesene	SH	0.101	±	0.03	0.058	±	0.05	0.143	±	0	0.167	±	0	RI, GC-MS	GC-MS
1459	1454	α-Humulene	SH	2.572	±	0.22	2.52	±	0.07	3.7	±	0.02	3.465	±	0.08	RI, GC-MS, Std	GC-MS, Std
-	1475	γ-Gurjunene	SH	-	-	-	0.9	±	0.03	-	-	-	-	-	-	-	GC-MS
1493	1485	β-Selinene	SH	3.653	±	0.33	3.358	±	0.15	5.242	±	0.14	4.757	±	0.11	RI, GC-MS	GC-MS
1486	1492	Valencene	SH	0.136	±	0.01	-	-	-	0.224	±	0.01	-	-	-	RI, GC-MS	-
1501	1494	α-Selinene	SH	2.409	±	0.23	1.972	±	0.29	3.493	±	0.08	3.009	±	0.09	RI, GC-MS	GC-MS
1510	1506	β-Bisabolene	SH	1.131	±	0.1	0.887	±	0.1	1.711	±	0.06	1.284	±	0.05	RI, GC-MS	GC-MS
1524	1520	7-epi-α-Selinene	SH	0.114	±	0.01	-	-	-	0.168	±	0.03	-	-	-	RI, GC-MS	-
-	1528	Calamenene	SH	-	-	-	-	-	-	-	-	-	0.013	±	0	-	GC-MS
1533	1529	γ-Bisabolene	SH	0.069	±	0.01	-	-	-	-	-	-	-	-	-	RI, GC-MS	-
-	1561	Nerolidol	SO	-	-	-	-	-	-	-	-	-	0.076	±	0	-	GC-MS
-	1577	Spathulenol	SO	-	-	-	0.184	±	0	-	-	-	0.118	±	0.09	-	GC-MS
-	1579	Isoaromadendrene epoxide	SO	-	-	-	0.086	±	0	-	-	-	-	-	-	-	GC-MS
1593	1582	Caryophylene oxide	SO	2.941	±	0.24	3.295	±	0.26	2.036	±	1.01	3.013	±	0.2	RI, GC-MS	GC-MS
1621	1608	Humulene epoxide II	SO	0.154	±	0.02	0.158	±	0	0.128	±	0.01	0.139	±	0	RI, GC-MS	GC-MS
1643	1638	Isospathulenol	SO	0.492	±	0.05	0.421	±	0.32	0.463	±	0.02	0.526	±	0.08	RI, GC-MS	GC-MS
1668	1651	Pogostole	SO	0.154	±	0.02	0.215	±	0.1	-	-	-	0.09	±	0	-	GC-MS
-	1658	Neointermedeol	SO	-	-	-	0.054	±	0.01	-	-	-	-	-	-	-	GC-MS
1675	1665	Intermedeol	SO	0.05	±	0.02	0.07	±	0.01	-	-	-	-	-	-	RI, GC-MS	GC-MS
1950	1938	Pellitorine	O	-	-	-	0.006	±	0.01	1.191	±	0.09	1.669	±	0.06	RI, GC-MS	GC-MS
-	1953	Hexadec-9-enoic acid	FAD	-	-	-	-	-	-	-	-	-	0.103	±	0	-	GC-MS
-	1959	Palmitic acid	FAD	-	-	-	-	-	-	-	-	-	0.423	±	0.04	-	GC-MS
-	2141	Oleic Acid	FAD	-	-	-	-	-	-	-	-	-	0.267	±	0.03	-	GC-MS
2707	2707	β-Monolinolein	E	-	-	-	-	-	-	1.84	±	0.98	-	-	-	RI, GC-MS	-
2018	NA	Heptadec-14-enal	A	-	-	-	-	-	-	0.122	±	0.04	-	-	-	RI, GC-MS	-
2815	NA	β-Monoolein	E	-	-	-	-	-	-	0.503	±	0.6	0.52	±	0.06	RI, GC-MS	GC-MS
-	NA	Hexadec-9-en-1-ol	O	-	-	-	-	-	-	-	-	-	0.238	±	0.01	-	GC-MS
-	2119 ^a^	17-Octadecynoic acid	FAD	-	-	-	-	-	-	-	-	-	0.025	±	0.02	-	GC-MS
-	NA	Tetradec-9-enal	A	-	-	-	-	-	-	-	-	-	0.073	±	0.01	-	GC-MS
-	2153 ^b^	Kalecide	O	-	-	-	-	-	-	-	-	-	0.181	±	0	-	GC-MS
-	2351 ^b^	Ricinoleic acid	FAD	-	-	-	-	-	-	-	-	-	0.231	±	0.01	-	GC-MS
-	1143 ^b^	Sabinol	SO	-	-	-	0.072	±	0.01	-	-	-	-	-	-	-	GC-MS
		Total identified (%)		99.4	±	0.2	99.2	±	0.1	98.7	±	0.7	96.7	±	0.1		

RI = retention indices for HP-5 column; Obs = retention indices determined relative to a homologous series of n-alkanes (C_8_–C_40_) on a HP-5MS column; Lit = literature RI values [41], ^a^ [45] (Tretyakov, 2007), ^b^ [38] (Andrianamaharavo, 2014); NA = RI values were not available in the literature; ^b^ C = Class. A—Aldehydes; E—Esters; FAD—Fatty acid and fatty acid derivatives; MH—Monoterpene hydrocarbons; MO—Oxygenated monoterpenes; O—Others; SH—Sesquiterpene hydrocarbons; SO—Oxygenated sesquiterpenes; Identification method: GC-MS = Mass spectrum was identical to that of the National Institute of Standards and Technology Library (ver. 2.0.f); RI = the retention index was matching the literature database; Std = constituent identity confirmed by co-injection of authentic standards; Retention indices were not calculated for compounds calculated only by the DB-HeavyWAX column; ^c^ When a hyphen “-” appears in the table in RI observed, it means it was a compound detected only by a DB-Wax column, where RI were not determined and calculated.

## Data Availability

All data are contained within the article.

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
