# Peer review of "Chemical Composition of Essential Oils and Supercritical Carbon Dioxide Extracts from Amomum kravanh, Citrus hystrix and Piper nigrum ‘Kampot’"

_molecules, 2023, doi:10.3390/molecules28237748_

Round 1

Reviewer 1 Report

Comments and Suggestions for Authors

The manuscript compares the chemical composition of the essential oils and supercritical CO2

extracts from the Cambodian spices: Amomum kravanh, Citrus hystrix, and Piper nigrum. The topic is interesting. However, I have a few observations:

1.       I recommend rereading the text for typos and grammatical mistakes.

2.       Add more details about the major findings of the study in the abstract.

3.       The introduction is lengthy and lacks a structure. Consider rewriting concisely.

4.       Aim: clarify the purpose of comparing the essential oils to the CO2 extracts. Is there a specific application?  

5.       Methods: What was the rate of distillation?

6.       Methods: How long was the CO2 extraction?

7.       Methods: Why did the authors use two types of GC columns?

8.       Rewrite and consolidate the conclusions in a more concise way. Focus on the major findings.

Comments on the Quality of English Language

I recommend rereading the text for typos and grammatical mistakes.

Author Response

November 16th, 2023

Dear Reviewer,

We have duly considered the suggestions on our manuscript Molecules-2705620 entitled Chemical composition of essential oils and supercritical carbon dioxide extracts from Amomum kravanh, Citrus hystrix and Piper nigrum 'Kampot' (authors: Vihanova K, Urbanova K, Nguon S and Kokoska L), which we find very valuable and acceptable. We have made the appropriate amendments of the manuscript and enclosed detailed replies to the reviewers’ comments (please see the attachment).

Yours sincerely,

Prof. Ladislav Kokoska

Department of Crop Sciences and Agroforestry

Faculty of Tropical AgriSciences

Czech University of Life Sciences Prague

Kamycka 129

165 21 Prague 6-Suchdol

Czech Republic

Reviewer 2 Report

Comments and Suggestions for Authors

Manuscript molecules-2705620 presents a GC-MS evaluation of the essential oils and CO2 extracts of three spices from Cambodia. Publication is recommended after addressing some minor points.

1.       Introduction: Please start new paragraphs at line 60 (To overcome…), line 99 (Leaves and fruits…), line 108 (Plants belonging…). Line 36, “Piperaceae” should not be italicized.

2.       Since a chiral GC-MS analysis was not carried out, the particular enantiomers of limonene and terpinen-4-ol cannot be determined. Please remove the “D” and “L” from these components.

3.       o-Cymene, identified in all three spices, may actually be p-cymene. See Romanenko & Tkachev, Chemistry of Natural Compounds, 2006, 42(6), 699-701. p-Cymene, RI = 1024; m-Cymene, RI = 1022, o-Cymene, RI = 1039. This makes more sense from a biosynthetic perspective; there are high concentrations of menthane monoterpenoids in the essential oils.

4.       “Humulene” should be α-Humulene

5.       To be consistent, use the (Z)/(E) nomenclature for orientation around a C=C double bond, use the cis/trans nomenclature for relative positions on a ring.

6.       There are several components listed where the literature RI values are “NA”. Check the NIST website (https://webbook.nist.gov/chemistry/) for RI values on columns of comparable polarity.

7.       Table 1:

a.       Isoterpinolene is a monoterpene hydrocarbon (MH)

b.       Eugenol is a phenylpropanoid and not an oxygenated monoterpenoid.

c.       Tetracosanol and octacosanol are alcohols and not aldehydes.

d.       The terms “K”, “AH”, “TH” should be defined in the footnote.

8.       Table 2

a.       m-Cymen-8-ol may actually be p-cymen-8-ol. Please double-check.

b.       cis-Ascaridol glycol can be classified as an oxygenated monoterpenoid (MO).

c.       “Germacrene D” rather than “D-Germacrene”.

d.       Although Adams lists an RI value for phytol as 1492, that is not correct. Many investigators find phytol with an RI of around 2100. RI 1942 is most likely isophytol.

e.       τ-Muurolol rather than tau.-Muurolol (i.e., use the Greek letter). τ-Muurolol = epi-α-Muurolol (Adams RI = 1640).

9.       Black pepper discussion: How does the composition compare with the ISO guidelines for P. nigrum EO?

Author Response

November 16th, 2023

Dear Reviewer,

We have duly considered the suggestions on our manuscript Molecules-2705620 entitled Chemical composition of essential oils and supercritical carbon dioxide extracts from Amomum kravanh, Citrus hystrix and Piper nigrum 'Kampot' (authors: Vihanova K, Urbanova K, Nguon S and Kokoska L), which we find extremely valuable and acceptable. We have made the appropriate amendments of the manuscript and enclosed detailed replies to the reviewers’ comments (please see the attachment).

Yours sincerely,

Prof. Ladislav Kokoska

Department of Crop Sciences and Agroforestry

Faculty of Tropical AgriSciences

Czech University of Life Sciences Prague

Kamycka 129

165 21 Prague 6-Suchdol

Czech Republic

Reviewer 3 Report

Comments and Suggestions for Authors

A summary and broad comments: The results can be of interest to the readers. The manuscript requires minor adjustments before publication (see comments).

Specific comments:

1) Title suggestion: Including Species Names in the Title

2) Line 125: the text reads, "In CO2 extracts, a total number 125 of 32, 36 and 31 components have been determined amounting to 98.65, 99.36 and 92.69 % 126 of the total extracts." It is not clear whether the results concern the analysis with an HP-122 5 column. Therefore, it is suggested to add this information to the text.

3) In line 248, "cold expression" should be corrected to "cold pressing”.

Author Response

(The authors gave the same response as above.)

Round 2

Reviewer 1 Report

Comments and Suggestions for Authors

The authors have made substantial improvements in the revised manuscript. I think it is ready for publication.